# Expansion and contraction of the flowing stream network alter hillslope flowpath lengths and the shape of the travel time distribution

5 H.J. Ilja van Meerveld[1], James W. Kirchner[2,3,4], Marc J.P. Vis[1], Rick S. Assendelft[1], Jan Seibert[1,5]

[1] Dept. of Geography, University of Zurich, Winterthurerstrasse 190, 8057 Zurich, Switzerland
[2] Dept. of Environmental System Sciences, ETH Zurich, 8092 Zurich, Switzerland
[3] Swiss Federal Research Institute WSL, 8903 Birmensdorf, Switzerland
[4] Dept. of Earth and Planetary Science, University of California, Berkeley, CA, 94720 USA
10 [5] Dept. of Aquatic Sciences and Assessment, Swedish University of Agricultural Sciences, P.O. Box 7050, 75007 Uppsala, Sweden

*Correspondence to*: Ilja van Meerveld (ilja.vanmeerveld@geo.uzh.ch)

**Abstract.** Flowing stream networks dynamically extend and retract, both seasonally and in response to precipitation events. These network dynamics can dramatically alter the drainage density, and thus the length of subsurface flow pathways to 15 flowing streams. We mapped flowing stream networks in a small Swiss headwater catchment during different wetness conditions and estimated their effects on the distribution of travel times to the catchment outlet. For each point in the catchment, we determined the subsurface transport distance to the flowing stream based on the surface topography, and determined the surface transport distance along the flowing stream to the outlet. We combined the distributions of these travel distances with assumed surface and subsurface flow velocities to estimate the distribution of travel times to the outlet. 20 These calculations show that the extension and retraction of the stream network can substantially change the mean travel time and the shape of the travel time distribution. During wet conditions with a fully extended flowing stream network, the travel time distribution was strongly skewed to short travel times, but as the network retracted during dry conditions, the distribution of the travel times became more uniform. Stream network dynamics are widely ignored in catchment models, but our results show that they need to be taken into account when modeling solute transport and interpreting travel time 25 distributions.

## 1. Introduction

Flowing stream networks extend and retract seasonally and during rainfall events (Ågren et al., 2015; Day, 1978; Gregory and Walling, 1968; Jensen et al., 2017; Peirce and Lindsay, 2015; Shaw, 2016). Some networks are less dynamic than others, depending on their geological and topographic settings (e.g., Whiting and Godsey, 2016), but many stream networks that are 30 not strongly controlled by persistent springs expand dramatically with increasing wetness conditions and streamflow. For example, the length of the flowing stream network in Sagehen Creek in California was 35 km during wet conditions in April

2008 but only 15 km during dry conditions in September 2006 (Godsey and Kirchner, 2014). The flowing stream drainage density of the completely extended stream network for a British peatland catchment was 20 times greater than that of the fully retracted stream network (Goulsbra et al., 2014). In an agricultural catchment in Oregon the flowing drainage density increased by two orders of magnitude between dry summer periods and wet winter periods (Wigington et al., 2005).

The expansion of the flowing stream network during wet periods increases the connectivity between hillslopes and streams. Wigington et al. (2005) argued that this increase in connectivity leads to higher nitrate exports because riparian buffer strips are largely bypassed, and travel times are shorter, when the flowing stream network is fully extended. Yet most catchment-scale solute transport studies assume static drainage networks, often derived from topographic maps that do not adequately represent intermittent streams. Even when intermittent streams are delineated as dashed lines on maps, their abundance is
often greatly underrepresented (Ågren et al., 2015; Brooks and Colburn, 2011; Fritz et al., 2013). Inadequate representation of the stream network can significantly impact the modeled retention capacity of riparian buffer strips (Baker et al., 2007) and thus solute export.

Travel time, i.e., the time it takes a raindrop to reach the catchment outlet, is an important control on the transport and fate of nutrients and contaminants, as well as mineral weathering. Because stream network expansion shortens the distances between hillslopes and flowing streams, it must also affect the distribution of travel times. However, most studies interpret
temporal variations in travel time distributions in terms of the relative contributions of fast and slow flow pathways and changes in the residence times of different storage zones, ignoring the effects of changes in the flowing stream network on subsurface flowpath lengths (Benettin et al., 2015a; Harman, 2014; van der Velde et al., 2012; Yang et al., 2018). Young water fractions were correlated with the drainage densities across 22 Swiss catchments, suggesting that denser drainage
networks, and thus shorter subsurface flowpaths, promote faster transport of recent precipitation (von Freyberg et al., 2018a). Hydrological modeling has similarly suggested a larger contribution of young water for lowland catchments with higher drainage densities and thus presumably shorter travel distances (Kaandorp et al., 2018).

Here, using simple graphical analyses of field-mapped stream networks, we show that network extension and retraction not only change subsurface travel distances and thus catchment-scale travel times, but also change the shape of the travel time
distribution. Our results imply that changes in the flowing stream network should be taken into account when modeling catchment-scale solute transport or interpreting travel time distributions.

## 2. Methods

### 2.1 Study site

For this study, we mapped flowing stream networks in a small headwater catchment in the Alptal, approximately 40 km
southeast of Zurich. Mean annual precipitation is 2300 mm y$^{-1}$, with roughly a third falling as snow (Stähli and Gustafsson,

2006). The wet climate and low-permeability Gleysols derived from Flysch bedrock (a sequence of sedimentary rocks, particularly argillite and bentonite schists, calcareous schists, marl and sandstone; Mohn et al., 2000; Schleppi et al., 1998) result in near-surface groundwater levels across much of the catchment (Rinderer et al., 2014). Streamflow generally responds very quickly (within tens of minutes) to rainfall. While most of the stormflow consists of pre-event water, event water contributions can be more than 50% (Fischer et al., 2017; von Freyberg et al., 2018b).

Our 13 ha headwater study catchment is located in the upper parts of the Studibach catchment and ranges in elevation from 1421 to 1656 m above sea level. The lower half of the catchment is forested, while the upper part is dominated by grasslands and wetlands that are used as meadows in summer (Figure 1). The average slope is 22°. In the lower part of the catchment, the stream is incised and the streambed contains large boulders; in the upper part of the catchment the streams are narrow (<0.2 m wide) and barely incised. For more information on the Studibach study catchment, see van Meerveld et al. (2017).

## 2.2 Stream networks used in this study

We manually surveyed the stream network by walking the entire catchment during different wetness conditions (including large events), using aerial photographs and GPS to ensure that the stream map included all streams. Our analysis uses the field-mapped flowing stream networks for three different dates with contrasting wetness conditions, as well as the complete network of all stream channels, which we assume represents the flowing stream network during extremely wet conditions. We mapped stream reaches with dry streambeds, pools of standing (but not flowing) water, or trickling flow conditions (<< 1 liter per minute based on visual observation) as dry channels. Even though the study area is generally very wet, the 2018 summer was extremely dry, leading to one of the lowest measured streamflows since 1968 in the neighbouring Erlenbach catchment. Field mapping during this period allowed us to obtain information about the minimum flowing stream length (Table 1). We assumed that the entire mapped channel network would be flowing during extremely wet conditions, although we never documented this situation because the stream network is very dynamic during rainfall events and field mapping is too slow to capture the maximum extent of the flowing stream network. We also compared our field-mapped networks to the stream network shown on the standard Swisstopo map (Federal Office of Topography, Swisstopo Pixelkarte 25; National Map 1:25,000; Figure 1). Thus, in total we compared five different flowing stream networks (Figure 2; Table 1):

1. Extremely dry conditions (Aug 21, 2018)
2. Dry conditions (Nov 2, 2016)
3. Wetting-up conditions (Oct. 25, 2016 during a low intensity rainfall event; 20 mm in total)
4. Complete network (assumed to represent the fully extended network during extremely wet conditions)
5. Topographic map (representing the stream network that would be assumed in the absence of field mapping)

The mapped flowing stream networks were significantly longer than the network shown on the Swisstopo map, except during the extremely dry conditions in August 2018 (Figure 2; Table 1). The flowing stream networks during the dry and wetting-up conditions in fall 2016 contained multiple dry sections in the steep central part of the catchment, separating the

upper parts of the flowing stream network from the outlet (Figure 2b-c). Such discontinuities in the flowing stream network have been observed in other catchments as well (e.g., Godsey and Kirchner, 2014; Whiting and Godsey, 2016)

## 2.3 Data analyses

Using the 2 m by 2 m LiDAR-derived digital elevation model for the catchment, we calculated the weighted mean length of all flow paths from each pixel to the nearest flowing stream pixel (with the weight based on the fraction of water taking each certain flowpath) based on the MD∞ algorithm (Seibert and McGlynn, 2007) (i.e., subsurface hillslope flow path length; $L_h$) and the travel distance through the flowing channel to the outlet ($L_s$) based on the D8 algorithm (O'Callaghan and Mark, 1984). For each pixel, we divided the average subsurface flow path length ($L_h$) by an assumed average subsurface velocity ($v_h$) to obtain an estimate of the subsurface travel time ($t_h$). We similarly divided the travel distance through the flowing stream channel ($L_s$) by an assumed average surface velocity ($v_s$) to obtain an estimate of the surface travel time ($t_s$). The subsurface and surface travel times were added to obtain an estimate of the total travel time to the catchment outlet (hereafter referred to as travel time; cf. Di Lazzaro (2009)) for each pixel ($t_t$):

$$t_t = t_h + t_s = \frac{L_h}{v_h} + \frac{L_s}{v_s}$$  eq. 1

We then determined the frequency distribution of the travel times ($t_t$) for all pixels in the catchment. This was done for each of the five stream networks. For all of the analyses shown here, we used 0.5 m s$^{-1}$ for the surface velocity ($v_s$) and 5 10$^{-4}$ m s$^{-1}$ for the subsurface velocity ($v_h$). Different subsurface velocities and surface to subsurface velocity ratios (from 10 to 10 000) were also tested. We also mapped the spatial distribution of pixels for which the estimated travel time was less than one or two days, assuming that these have the potential to contribute to stormflow.

These calculations include several subjective decisions and simplifying assumptions (i.e., that velocities are constant in space and time, that all areas in the catchment contribute equally to discharge at the outlet, and that the flowing stream network remains stable for long enough so that travel times at the outlet can be expressed as a static transit time distribution). Our main objective is to illustrate the effects of changes in the flowing stream network on subsurface flow path lengths and thus the travel time distributions. These effects are best illustrated by keeping all other factors constant, using the simplifying assumptions outlined above. Previous work (Mutzner et al., 2016) has shown how different methods to extract the channel network affect hillslope-to-stream travel distances (i.e., rescaled width functions) and thus the derived geomorphological instantaneous unit hydrograph. Here, our focus is not on the effects of different stream network extraction methods, but rather on how changes in the flowing stream network affect subsurface travel distances and catchment-scale travel times.

## 3. Results

Extension of the flowing stream network during wet conditions significantly shortens the subsurface flow pathways (shown in red for five selected locations A-E in Figure 3). This not only shortens the average and median travel time to the outlet, but also changes the shape of the travel time distribution (Table 2 and Figure 4a-d). For the extended flowing stream networks typical of wet conditions, most subsurface travel distances (and thus travel times) are short, but for the retracted networks typical of dry conditions, the travel times are longer and more uniformly distributed. When the flowing stream network is greatly retracted during extremely dry periods, almost the entire catchment has travel times longer than two days and thus could not contribute to stormflow in response to a brief rainfall event. However, when the flowing stream network is fully extended, most of the catchment could contribute to stormflow at the outlet because the travel times are mainly short (Figure 4d). The correspondence between flowing stream networks and travel time distributions is not one to one, however. For example, even though the flowing stream network during the dry conditions in November 2016 is different from the network shown on the topographic map (Figure 2b and e), the cumulative frequency distributions of the travel times are similar (Figure 5).

The travel time distribution for the stream network during the wetting-up period (October 2016 mapping) is bimodal (Figure 4c) due to the large area with flowing streams that is disconnected from the outlet by the dry stream section in the steeper part of the catchment (Table 1 and Figure 2c). For the selected subsurface velocity ($v_h$) of $5 \cdot 10^{-4}$ m s$^{-1}$, almost two days are required to cross the dry part of the channel as subsurface flow. A less apparent bi-modal travel time distribution also results from disconnection of the flowing stream network during the extremely dry conditions of August 2018 (Figure 4a).

The chosen surface and subsurface velocities do not substantially affect the shapes of the travel time distributions (Figure 6). Changing the assumed subsurface velocity (and thus the ratio of the surface to subsurface velocities) by large factors has the effect of rescaling the travel time distributions but does not substantially change their shapes (Figure 6). This is to be expected. The shapes of the travel time distributions will be mainly determined by the distribution of subsurface travel distances ($L_h$), whenever velocities are assumed to be constant in space and time, and slower in the subsurface than the surface. Under these assumptions, the subsurface travel times ($t_h$) will be much longer than the surface flow travel times ($t_s$), and thus will largely determine the travel time distribution. Reasonable ranges of assumed surface flow velocities have virtually no effect on the travel time distributions, due to the very small contribution of the surface flow travel times ($t_s$) to the total travel times ($t_t$).

## 4. Discussion

By only changing the flowing stream network and keeping all other variables (such as the velocities) constant, our analysis shows how the extension and retraction of the flowing stream network affect subsurface flowpath lengths and catchment-scale travel times. In practice, the effects of catchment wetness on travel time distributions will be larger than shown here,

because subsurface flow velocities will be smaller during dry conditions, significantly increasing travel times when the stream network is most contracted. Subsurface flow velocities will also vary spatially, which will further broaden the travel time distributions. Furthermore, subsurface flow directions may not follow the surface topography and may change depending on water table gradients and thus wetness conditions (Rodhe and Seibert, 2011; van Meerveld et al., 2015), and some areas of the catchment may not contribute to streamflow during dry conditions (Jencso et al., 2010; Zuecco et al., 2019). By excluding these confounding factors, we could isolate the effect of stream network geometry on travel times, and show that stream network extension and retraction significantly alter the mean and median travel times, as well as the shape of the travel time distribution.

Previous modeling studies have suggested that streamflow consists of a larger fraction of young water during wet conditions than during dry conditions. For example, Benettin *et al.* (2015b) calibrated a hydrological model for the Plynlimon catchment in Wales using both streamflow and stream chloride data, and suggested that the travel time distribution was much more skewed towards younger water during wet conditions. Visser *et al.* (2019) used a combination of isotope tracers to constrain a hydrological model for a Sierra Nevada catchment and inferred that the travel time distribution was skewed towards younger water during high-flow conditions but was nearly uniform during baseflow (although this was partly due to a lack of young water in storage due to drought conditions). This change in the streamflow travel time distribution (and the storage selection function) with catchment wetness conditions is generally attributed to a larger contribution from shallower and faster flow pathways during wetter conditions (Benettin et al., 2015b; Harman, 2014; Hrachowitz et al., 2016; van der Velde et al., 2012). Although the travel times in these studies were much longer than we calculated here, in part because we assumed that surface and subsurface flow velocities would not decrease during dry conditions, our results suggest that the dynamics of the flowing stream network alone can lead to significant changes in travel time distributions. Therefore, these network dynamics and the associated changes in subsurface travel distances need to be taken into account when interpreting time-varying travel time distributions. Above all, more studies are needed where detailed tracer sampling is combined with detailed stream network mapping to determine how stream network extension affects travel time distributions. Our results also suggest the speculative possibility that the dynamics of stream network extension and retraction could potentially be inferred from the time-varying behaviour of travel time distributions.

Our results, furthermore, suggest that stream networks shown on the topographic maps may loosely approximate the flowing stream network during dry conditions, but not during wet conditions. When these static networks are used for modeling studies, the modeled flow pathways may be far longer than the real-world subsurface flowpaths, particularly during wet conditions (see also Zimmer and McGlynn, 2018). The resulting modeled transit time distributions would then be much less skewed than those in the real world. This would lead to much slower modeled transport of pollutants, unless compensated otherwise (e.g. via unrealistically high velocities or large areas with surface runoff, as for example shown for flow on the Greenland ice sheet by Yang et al. (2018)). Therefore, solute transport studies need to take the complexities of stream network extension and retraction into account, particularly in locations where (or at times when) the network may be very

dynamic. This will require better knowledge of the processes and catchment characteristics that control flowing stream network extension and retraction, since it is impractical to map the dynamics of the flowing stream network in every catchment. As more field maps of network extension and retraction become available, empirical generalizations about stream network dynamics and their controlling factors will become more reliable. As an example of what may be possible, Prancevic and Kirchner (2019) have recently shown that topography may be a useful predictor of where the flowing stream network is highly dynamic and where it is more stable. Using either empirical generalizations from the limited available field studies, predictive relationships like those suggested by Prancevic and Kirchner (2019), or modeled stream networks (Russell et al., 2015; Ward et al., 2018; Williamson et al., 2015) would be better than assuming that flowing stream networks are static.

## 5. Conclusion

We estimated travel time distributions for different mapped stream networks by calculating the subsurface transport distance from each pixel to the nearest flowing stream and the surface transport distance along the stream network to the outlet for different flowing stream networks. Our results show that extension and retraction of flowing stream networks can significantly alter catchment travel time distributions, even if all other factors remain constant. When stream networks extend during wet conditions, travel times become shorter and their distributions become more skewed. Conversely, when stream networks retract during dry conditions, travel times become longer and more uniformly distributed. The effects of flowing stream network dynamics will be even more significant in the real world than calculated here, because we assumed that velocities did not change with wetness conditions, in order to isolate the effect of stream network geometry alone. Our simple graphical analysis implies that the dynamics of the flowing stream network need to be taken into account when interpreting travel time distributions or modeling solute transport. This will require better documentation of stream network extension and retraction in more diverse landscapes and climatic conditions, coupled with a better understanding of the processes and catchment characteristics that control flowing stream network dynamics.

## 6. Acknowledgements

We thank Oskar Sjöberg for his help with the initial surveys to create the stream map. This work was funded by the Swiss National Science Foundation (project STREAMEC; 159254).

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

| | | Mapped networks | | | | Topographic map |
|---|---|---|---|---|---|---|
| | | Extremely dry | Dry | Wetting-up | Complete network | |
| Streamflow | (mm d$^{-1}$) | 0.2 | 0.5 | 8.1 | - | - |
| | percentile | 96 | 82 | 18 | - | - |
| Flowing stream network length (km) | | 0.63 | 1.11 | 3.11 | 3.77 | 0.68 |
| Flowing stream network density (km km$^{-2}$) | | 4.9 | 8.5 | 23.9 | 29 | 5.2 |
| Connected flowing stream length (km) | | 0.42 | 0.39 | 1.57 | 3.4 | 0.68 |
| Fraction connected (-) | | 0.65 | 0.35 | 0.50 | 0.90 | 1 |

**Table 1. Flowing stream network length, flowing stream density, flowing stream length that was connected to the outlet, and the fraction of the flowing stream length that was connected to the outlet for the five stream networks used in this study. Daily streamflow at the neighbouring 70 ha Erlenbach catchment and the percentile of flow based on the 1978-2018 flow record, are given for comparison of the wetness conditions as well. Note that we assume that during extremely wet conditions flow occurs throughout the complete network but that we did not survey the network during these conditions. For the 1978-2018 flow record, the average annual maximum daily flow for the Erlenbach catchment was 67 mm d$^{-1}$, and the average daily flow was 4.8 mm d$^{-1}$.**

| | | Mapped stream networks | | | | Topographic map |
|---|---|---|---|---|---|---|
| | | Extremely dry | Dry | Wetting-up | Complete network | |
| Travel time | Mean (days) | 6.3 | 4.5 | 2.5 | 1.6 | 4.7 |
| | Median (days) | 6.5 | 4.1 | 2.5 | 1.0 | 4.5 |
| | Interquartile range (days) | 6.0 | 5.1 | 2.9 | 2.0 | 5.6 |
| | Skewness | -0.03 | 0.31 | 0.56 | 1.47 | 0.20 |
| Subsurface travel time | Median (days) | 6.5 | 4.1 | 2.4 | 1.0 | 4.5 |
| Surface travel time | Median (days) | $3.3 \ 10^{-3}$ | $5.7 \ 10^{-3}$ | $8.3 \ 10^{-3}$ | $9.3 \ 10^{-3}$ | $4.9 \ 10^{-3}$ |
| Fraction of catchment with travel time | ≤1 day (-) | 0.09 | 0.13 | 0.27 | 0.51 | 0.16 |
| | ≤2 days (-) | 0.15 | 0.26 | 0.43 | 0.71 | 0.26 |

**Table 2. Statistics for the travel time distributions ($t_t$), as well as the median subsurface ($t_h$) and surface ($t_s$) travel times, and the fraction of the catchment with travel times shorter than or equal to one or two days, for the five different stream networks using a subsurface velocity ($v_h$) of $5 \cdot 10^{-4}$ m s$^{-1}$ and surface velocity ($v_s$) of 0.5 m s$^{-1}$. See Figure 2 for the maps with the stream networks and**
5 **Figure 4 for the travel time distributions and maps of the areas with travel times shorter than one and two days.**

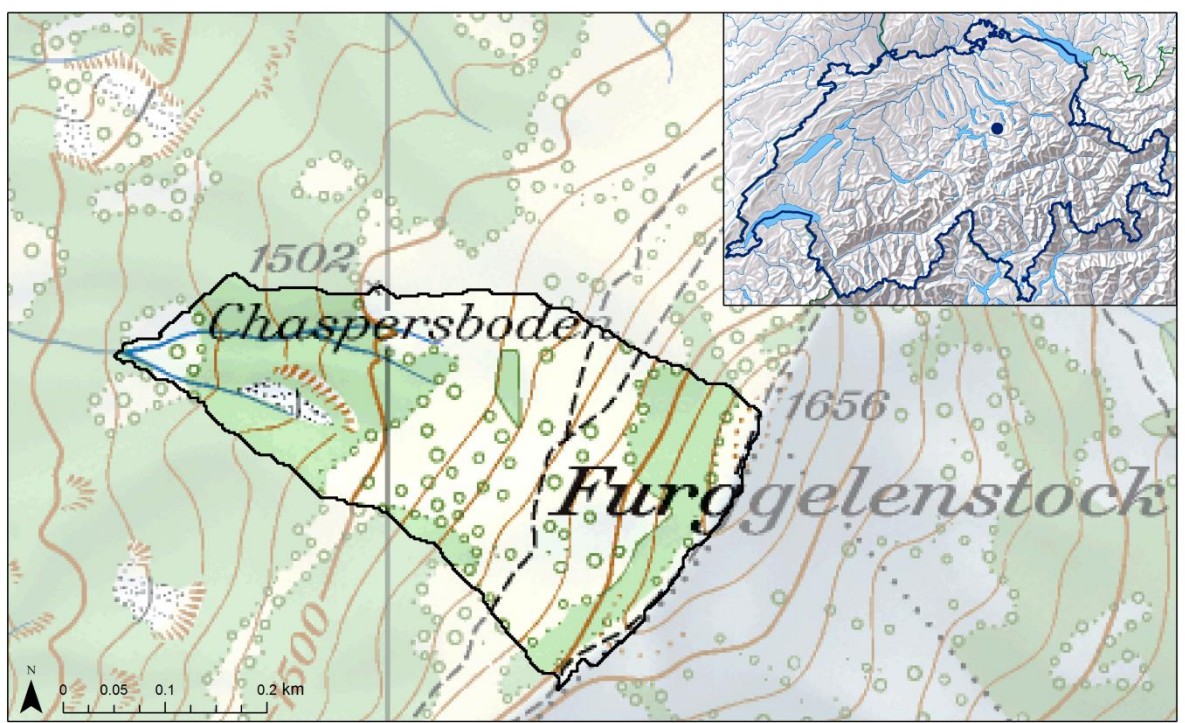

**Figure 1. Map of the upper Studibach study catchment and its location in Switzerland (inset). Source: Federal Office of Topography (Swisstopo) National Map 1:25,000 (Pixelkarte 25) and Reliefkarte 1:2,000,000.**

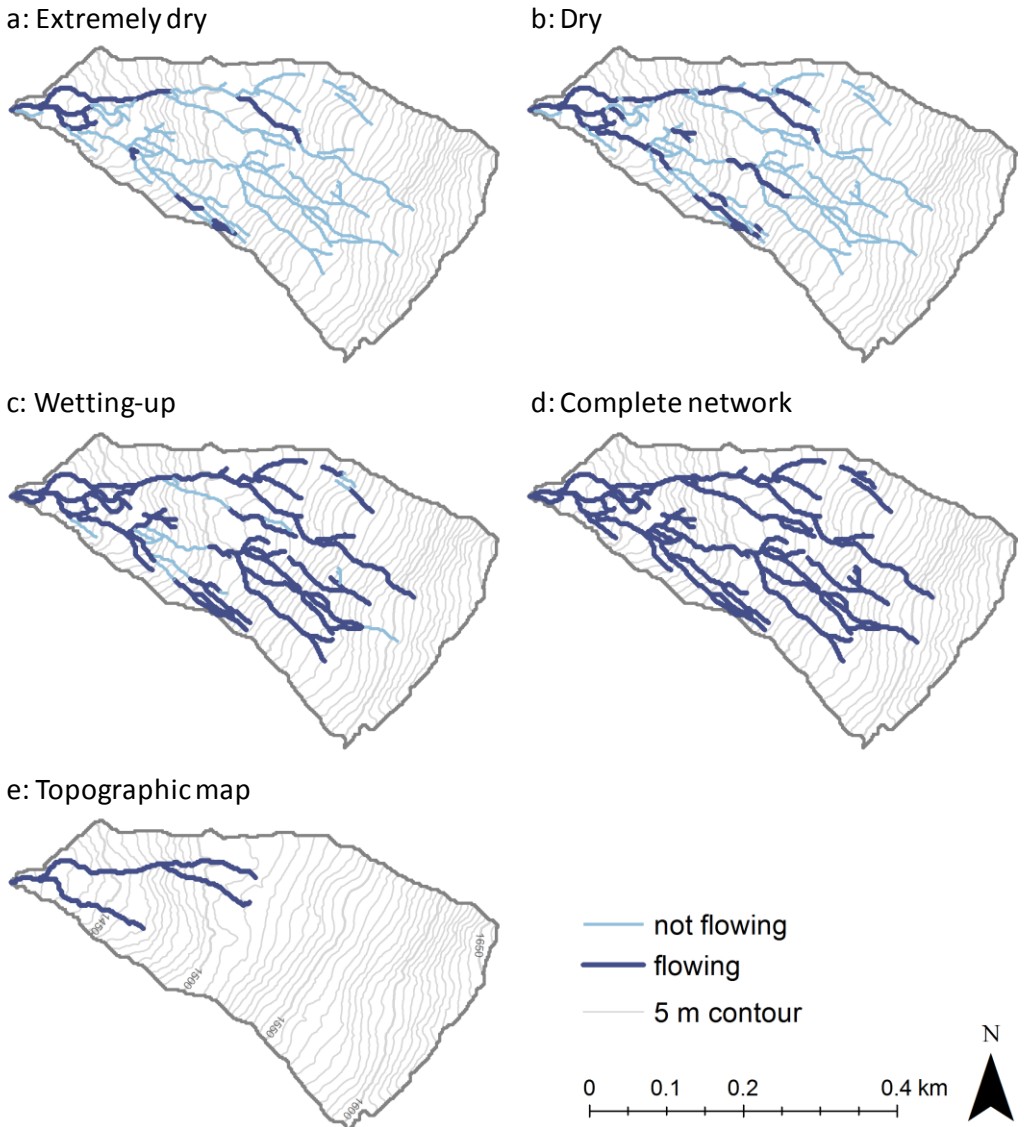

**Figure 2. Maps of the five stream networks (flowing in dark blue and not flowing in light blue) used in this study. a: extremely dry conditions observed on 21 August 2018; b: dry conditions observed on 2 November 2016; c: wetting-up conditions observed during a rainfall event on 25 October 2016; d: the complete stream network assumed to represent the flowing stream network during extremely wet conditions; e: the stream network shown on the 1:25,000 topographic map (see Figure 1). The length of the flowing stream network changes dramatically with wetness conditions and is significantly underrepresented by the stream network shown on the topographic map.**

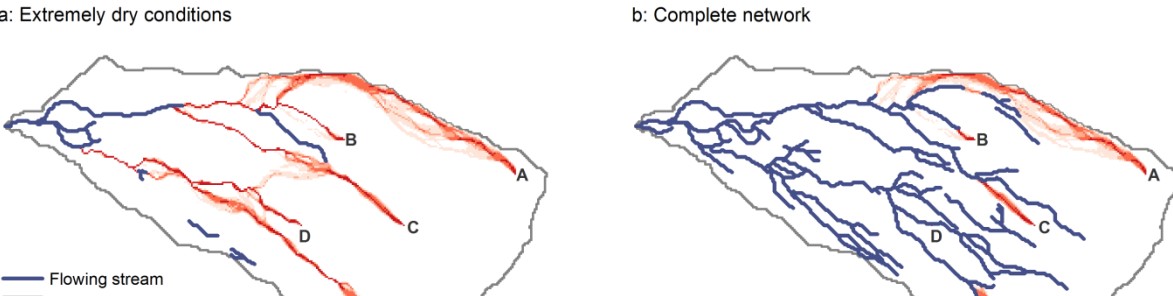

**Figure 3. Maps showing subsurface flow pathways starting from five selected pixels (in red; A-E) and the flowing stream network (in blue) observed during extremely dry conditions and for the complete network (assumed to represent extremely wet conditions). Darker colors indicate a larger fraction of the flow. The shorter flowing stream network under dry conditions implies much longer subsurface flow pathways from most points on the landscape. The subsurface fractions of the total travel distance to the outlet ($L_h/L_t$, m/m) for the extremely dry and complete network are: A: 0.66 and 0.44; B: 0.48 and 0.07; C: 0.59 and 0.15; D: 0.74 and 0.01; E: 0.81 and 0.11, respectively.**

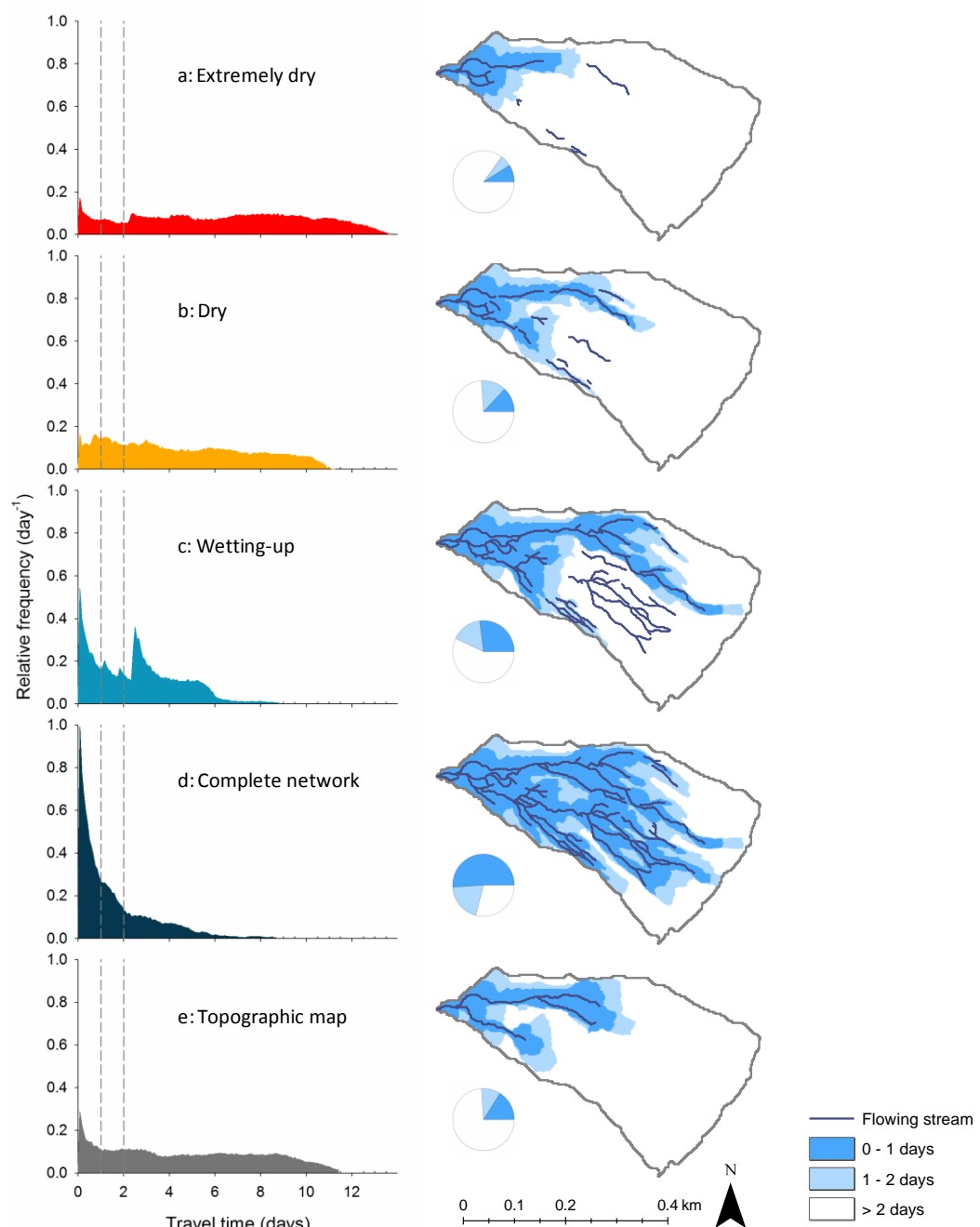

**Figure 4. Effects of flowing stream network extension and retraction on the travel time distributions.** The left hand column shows the distributions of travel times ($t_t$) to the catchment outlet for the five flowing stream networks. The right hand column shows the networks themselves, as well as the locations in the catchment with travel times ≤ 1 and 1-2 days (dark blue and light blue, respectively, corresponding to the fractions of catchment area shown in the pie charts). Travel times were calculated assuming a subsurface velocity ($v_h$) of $5 \cdot 10^{-4}$ m s$^{-1}$ and a surface velocity ($v_s$) of 0.5 m s$^{-1}$. See Table 2 for the main descriptive statistics of the travel time distributions. Under wetter conditions, more of the catchment area lies close to flowing streams; thus travel times are shorter and their distribution is more skewed.

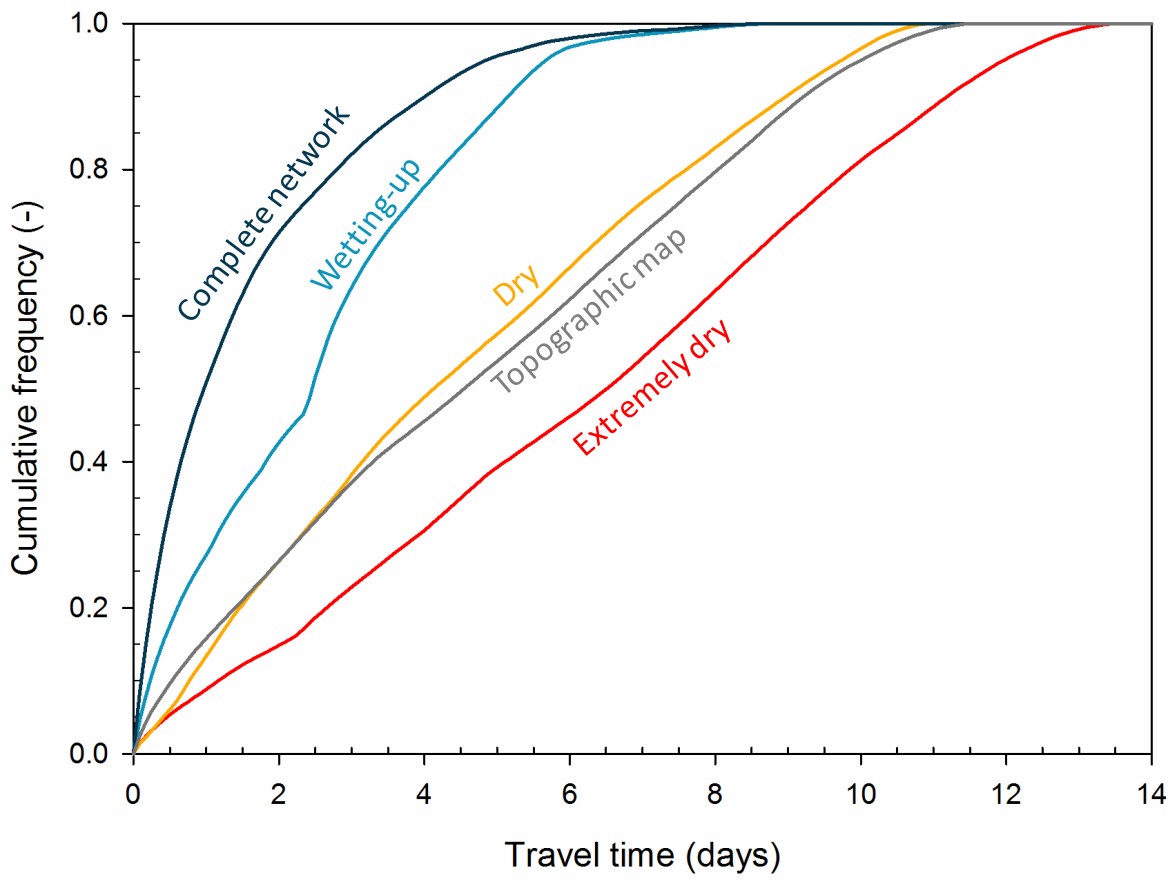

**Figure 5. Cumulative frequency distributions of the travel time ($t_t$) to the catchment outlet for the five flowing stream networks shown in Figures 2 and 4. See Table 2 for the main descriptive statistics of the travel time distributions.**

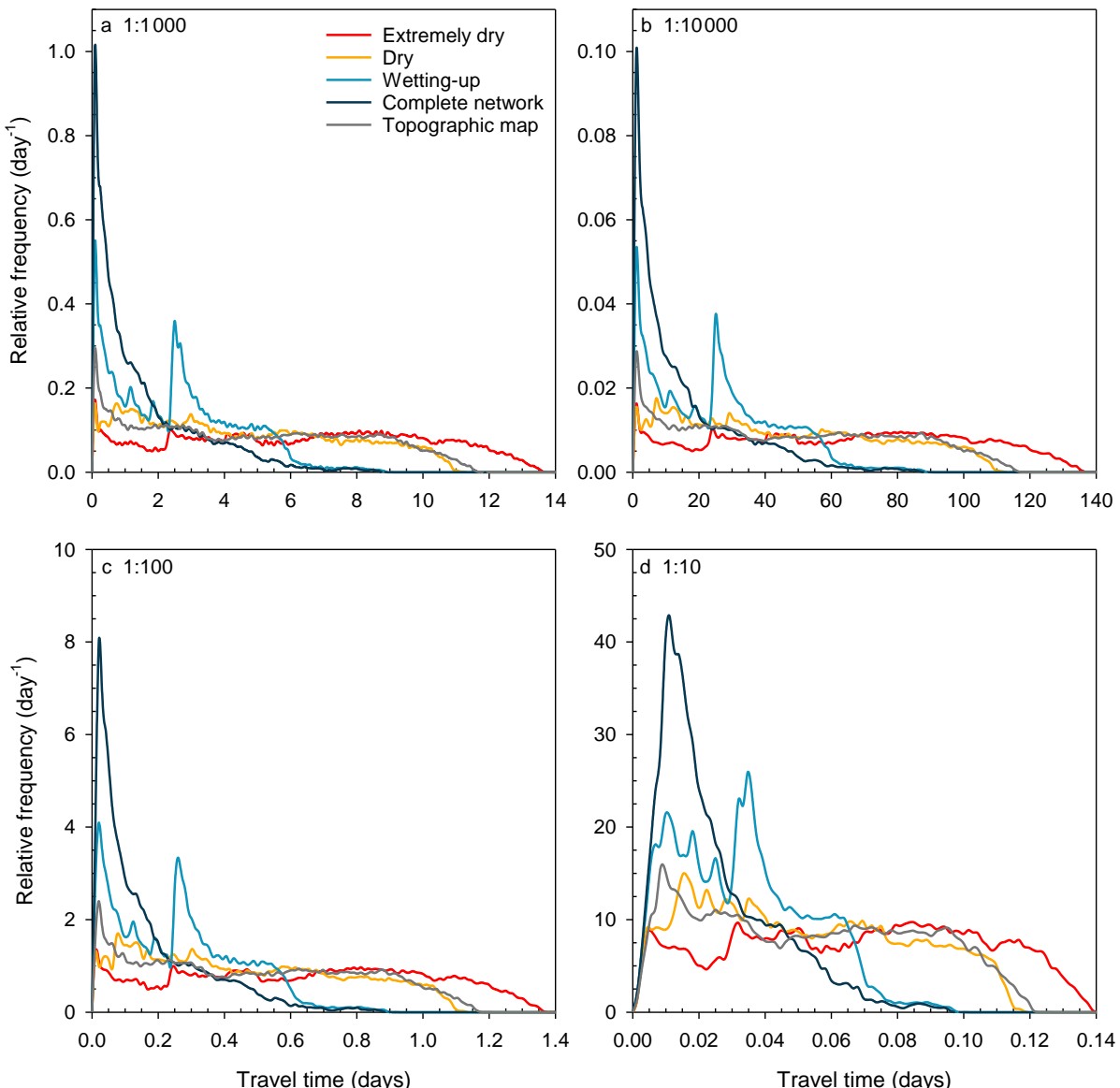

**Figure 6. Different assumed subsurface flow velocities change the travel times but not the shapes of their distributions.** The panels show the travel time distributions for the five flowing stream networks, assuming a surface velocity ($v_s$) of 0.5 m s$^{-1}$ and subsurface velocities ($v_h$) of a: $5 \cdot 10^{-4}$ m s$^{-1}$ (as used in Figure 4), b: $5 \cdot 10^{-5}$ m s$^{-1}$, c: $5 \cdot 10^{-3}$ m s$^{-1}$, and d: $5 \cdot 10^{-2}$ m s$^{-1}$. The value shown in the upper left corner of each panel represents the ratio of the subsurface to surface velocities ($v_h : v_s$).