# Peer review of "Expansion and contraction of the flowing stream network alter hillslope flowpath lengths and the shape of the travel time distribution"

_Hydrology and Earth System Sciences, 2019_

## Short Comment (SC1) · 30 May 2019

A very interesting study! It has been long known that hillslope and open-channel partition impacts catchment hydrographs but few studies have estimate this point using high-quality dataset. The main contribution of this study is to map multi-temporal, accurate flowing stream networks and investigate their impacts on catchment hydrographs. Some important implications have been reported as well.

This reminds me of the work I've done for routing surface meltwater on the Greenland

ice sheet. I've found hillslope and open-channel partition impacts surface meltwater discharge at the catchment outlet. However, I used a series of cumulative area thresholds to create dynamic supraglacial stream networks from DEMs (see Figure 7). This study has done a better work: instead of using DEM simulations, real field-measured stream networks are used.

If the authors are interested, see my paper published in The Cryosphere: Yang, K., Smith, L.C., Karlstrom, L., Cooper, M.G., Tedesco, M., As, D.v., Cheng, X., Chen, Z., Li, M., 2018. A new surface meltwater routing model for use on the Greenland Ice Sheet surface. Cryosph. 12, 3791-3811. https://www.the-cryosphere.net/12/3791/2018/

---

## Referee Comment (RC1) · Anonymous Referee #1 · 25 Jun 2019

Expansion and contraction of the flowing stream network changes hillslope flowpath lengths and the shape of the travel time distribution van Meerveld et al.

General comments

This article presents an interesting thought experiment about how riverine network length can influence the mean travel time distribution in catchments. The authors present a set of feasible river network extents across a range of wetness conditions, assume surface and subsurface flow velocities, and then estimate plausible distributions of travel times to the catchment outlet within these wetness scenarios. As this study is an initial exploration of how network extent can influence travel time distributions and modeling solute transport, I believe this study would be more powerful if the authors emphasized how future studies can build off of this initial exploration. For instance, emphasizing what the limitations of this study design are, and how others can use these concepts and apply real datasets and hydrologic measurements to confirm the results and interpretations of this study, would be greatly beneficial. Since this study estimates subsurface and surface velocities, it seems appropriate to provide results from a sensitivity analysis or provide ranges in the mean travel times. While the authors state they tested surface to subsurface velocity ratios (from 10 to 10000; P 4 L 14), they do not appear to present the results of that analysis. A powerful addition to this paper would be to show possible ranges in mean transit time distributions, given minimum and maximum velocities.

Specific comments

1. P 3 L 26: How substantial of a rainfall event? Is the rainfall occurring in "wet" conditions? I suspect not as it occurs right before the "dry" conditions survey.

2. P 3 L 20: The authors say that the field mapping is too slow during rainfall events to capture the entire extent of the stream work during rainfall events due to how dynamic it is. However, it appears the authors use a survey taken during a rainfall event in this analyses. Thus, it would be helpful to the reader if more information was provided on these surveys, e.g. how long did the surveys take, did the researchers start at the channel heads and walk down (to ensure they capture the most dynamic extents), was the network actively expanding during the survey, etc?

3. P 3 L 27: It may be more clear to the reader how survey #4 was accomplished (every other survey is described in parentheses, but this one).

4. There have been several recent studies that sought to predict river network extent, which can be used to model transit time distributions as suggested by the authors on

P33. Some suggestions below for two recent studies that can be used:

P 7 L 1: Add another example of predictive modelling: Ward, A. S., Schmadel, N. M., & Wondzell, S. M. (2018). Simulation of dynamic expansion, contraction, and connectivity in a mountain stream network. Advances in Water Resources, 114, 64-82.

P 6 L 33- P 7 L 1: Add example of empirical generalization from field studies, such as: Zimmer, M. A., & McGlynn, B. L. (2018). Lateral, vertical, and longitudinal source area connectivity drive runoff and carbon export across watershed scales. Water Resources Research, 54(3), 1576-1598.

This study also relates network expansion and retraction to solute transport dynamics as well, which is suggested in this study, but few if any citations are provided.

5. TABLE 1: While it is clear why the topographic map does not have an associated streamflow, please add brief explanation in caption as to why streamflow magnitude is not provided for complete network.

6. TABLE 2: This is an incredibly interesting results table and definitely made me think about possible travel times in other catchments and across wetness conditions. While I think the authors main points from this paper were to show that travel times decrease substantially as the system wets up, the absolute values for the reported median travel times are very small. The median surface travel times are on the order of minutes – how did the authors determine this? Based on the catchment and previous field observations, does it seem reasonable that 71% of the water travel time are less than 2 days?

7. It is also interesting that the median travel time for the topographic map survey is 4.5 days and the subsurface travel time is 4.5 days, which are both longer than the "dry" conditions survey, and yet the fraction of the catchment with travel time less than 1 day is greater for the topographic map. Perhaps this is driven by the hydrologic connectivity of the river network in the topographic map survey. This is an interesting dynamic that

could be expanded on in this paper and could be related to recent papers on the topic of discontinuous network extents, such as:

Godsey, S. E., & Kirchner, J. W. (2014). Dynamic, discontinuous stream networks: hydrologically driven variations in active drainage density, flowing channels and stream order. Hydrological Processes, 28(23), 5791-5803.

Whiting, J. A., & Godsey, S. E. (2016). Discontinuous headwater stream networks with stable flowheads, Salmon River basin, Idaho. Hydrological Processes, 30(13), 2305-2316.

8. FIGURE 2: What is the role of disconnected stream channels in the model results? Do water parcels flow through these disconnected sections at the same rate as those coming from the terrestrial landscape outside the channel extent? Do the authors think that subsurface flow map be faster within the subsurface channel network than in the hillslopes adjacent to the network?

Technical corrections and editorial suggestions

P 5 L 11: missing "x" between "5" and "10". P 6 L 13: Delete "did" at end of sentence. TABLE 2: Change "travel times smaller than one and two days" to "travel times shorter than one and two days"
* * *

---

## Author Comment (AC1) · 9 Aug 2019

**Response to reviewer 1 comments**
We appreciate the comments by the reviewer and the general positive assessment of the manuscript. Below we respond (in blue text) to the individual comments (in black text).The comments will help us to clarify the manuscript. In some cases the reviewer asks for information/discussion that we already provide in the manuscript. In these cases, our response explains where we provided this information/discussion, but of course we can extend these parts if the reviewer/editor thinks that additional text is needed.

General comments
This article presents an interesting thought experiment about how riverine network length can influence the mean travel time distribution in catchments. The authors present a set of feasible river network extents across a range of wetness conditions, assume surface and subsurface flow velocities, and then estimate plausible distributions of travel times to the catchment outlet within these wetness scenarios. As this study is an initial exploration of how network extent can influence travel time distributions and modeling solute transport, I believe this study would be more powerful if the authors emphasized how future studies can build off of this initial exploration. For instance, emphasizing what the limitations of this study design are, and how others can use these concepts and apply real datasets and hydrologic measurements to confirm the results and interpretations of this study, would be greatly beneficial.

> We describe the limitations of the study (particularly the uniform and constant velocities and the 'steady state assumption' for each stream network) on P5L24-31 (first part of the discussion). Our main goal was to show that the geometry of the flowing stream network affects the distribution of the hillslope travel distances to the flowing streams (and to a much smaller extent the travel distances in the stream) and thus the travel time distribution. We describe in the discussion what these results mean for interpreting travel time distributions obtained from tracer data and highlight that these results should be considered in solute transport models that - so far - tend to use a fixed (rather than dynamic) stream network.

Since this study estimates subsurface and surface velocities, it seems appropriate to provide results from a sensitivity analysis or provide ranges in the mean travel times. While the authors state they tested surface to subsurface velocity ratios (from 10 to 10000; P 4 L 14), they do not appear to present the results of that analysis. A powerful addition to this paper would be to show possible ranges in mean transit time distributions, given minimum and maximum velocities.

> We actually provide the results for different velocities in Figure 6 and describe them in the last paragraph of section 3. In short, the chosen velocities greatly affect the mean travel times and the range of travel times but have a minor effect on the shape of the travel time distribution or the differences in the travel time distributions for the different stream networks. Thus the main result of this study (namely, that the geometry of the flowing stream network affects the shape of the travel time distributions) does not depend on our chosen velocities. See also the response to specific comment 6 below.

Specific comments
1. P 3 L 26: How substantial of a rainfall event? Is the rainfall occurring in "wet" conditions? I suspect not as it occurs right before the "dry" conditions survey.

> There were 27 mm of precipitation on October 25[th] and another 31 mm fell on October 26th. There was no other rain after this, even until November 2[nd] (total rainfall between October 15[th] and November 2[nd] was 83 mm). Streamflow in the catchment responds quickly to precipitation (within minutes to hours) and baseflow is generally reached within one to two days after an event. Thus, by November 2[nd], streamflow had returned to baseflow conditions, although the lowest flows during this fall period were increasing slightly (see Figure R1 below).

[Figure]

Figure R1. Daily precipitation and streamflow at the Erlenbach gauging station during spring to-fall 2016. The red lines indicate the times of the two stream surveys in fall 2016. Note the log-scale for streamflow. The flow during the extremely dry conditions in summer 2018 was 0.18 mm/d. The data were obtained from Stähli (2018), Long-term hydrological observatory Alptal (central Switzerland); https://www.envidat.ch/dataset/longterm-hydrological-observatory-alptal-central-switzerland.

2. P 3 L 20: The authors say that the field mapping is too slow during rainfall events to capture the entire extent of the stream work during rainfall events due to how dynamic it is. However, it appears the authors use a survey taken during a rainfall event in this analysis. Thus, it would be helpful to the reader if more information was provided on these surveys, e.g. how long did the surveys take, did the researchers start at the channel heads and walk down (to ensure they capture the most dynamic extents), was the network actively expanding during the survey, etc?

We didn't survey the stream starting at the channel heads but rather walked along the contours and then up the different streams (thus the surveys were done in more of a zigzag pattern across the catchment). Each survey took at least half a day to complete. Since the peak of the event is very short, we cannot survey the entire stream network at the peak of the event. We will make it clearer in the text of the manuscript that the mapping is too slow to capture the peak flow conditions during an event.

The October 2016 stream network was mapped in the afternoon of October 25th during an event with low intensity rainfall (see Figure R2 below). Total rainfall was 10 mm by noon (when the mapping started), 16 mm by 3 pm, and 20 mm at 5 pm when the mapping was completed.

[Figure]

Figure R2. Hourly and cumulative precipitation recorded at the lower and upper rain gauge in the Studibach before and during the stream survey on October 25, 2016.

3. P 3 L 27: It may be more clear to the reader how survey #4 was accomplished (every other survey is described in parentheses, but this one).

> The stream network was surveyed on multiple occasions to ensure that we had mapped all streams. The complete network is assumed to represent the fully extended network during extremely wet or peak flow conditions. We did not observe flow in all streams at the same time; instead, #4 represents a hypothetical scenario in which all channels are flowing during extremely wet conditions. We explained this in the text but will try to make it clearer. We will move this clarifying text to the parentheses and make sure that in the figure and table caption it is clear that the complete network is assumed to represent the flowing stream network during extremely wet conditions.

4. There have been several recent studies that sought to predict river network extent, which can be used to model transit time distributions as suggested by the authors on
P33. Some suggestions below for two recent studies that can be used:
P 7 L 1: Add another example of predictive modelling: Ward, A. S., Schmadel, N. M., & Wondzell, S. M. (2018). Simulation of dynamic expansion, contraction, and connectivity in a mountain stream network. Advances in Water Resources, 114, 64-82.
P 6 L 33- P 7 L 1: Add example of empirical generalization from field studies, such as: Zimmer, M. A., & McGlynn, B. L. (2018). Lateral, vertical, and longitudinal source area connectivity drive runoff and carbon export across watershed scales. Water Resources Research, 54(3), 1576-1598.
This study also relates network expansion and retraction to solute transport dynamics as well, which is suggested in this study, but few if any citations are provided.

> Thank you for these suggestions. We will add references to these papers and other stream network model studies to the manuscript.

5. TABLE 1: While it is clear why the topographic map does not have an associated streamflow, please add brief explanation in caption as to why streamflow magnitude is not provided for complete network.

> We will add the clarification to the table that we never observed flow in the complete stream network but that we assume that this is the case for peak flow conditions during very large events.

6. TABLE 2: This is an incredibly interesting results table and definitely made me think about possible travel times in other catchments and across wetness conditions. While I think the authors main

points from this paper were to show that travel times decrease substantially as the system wets up, the absolute values for the reported median travel times are very small. The median surface travel times are on the order of minutes – how did the authors determine this? Based on the catchment and previous field observations, does it seem reasonable that 71% of the water travel time are less than 2 days?

> The main point of the study was indeed to show that the changes in the flowing stream network geometry can significantly affect the travel time distribution. We did not intend to derive actual travel time distributions for this catchment but rather focus on how the distributions differ between the different stream networks.
>
> We don't know the average surface and subsurface velocities in the catchment and therefore state clearly that these are assumed values. We test the effect of using different velocities and show that the effect of the chosen velocities on the shape of the travel time distribution is minimal (see Figure 6 in the manuscript). Furthermore, we discuss on P5L4-31 the implications of using uniform and constant velocities.
>
> The average surface velocity of 0.5 m/s is typical for mountain streams. The average subsurface velocity of $5 \times 10^{-4}$ m/s is high compared to the hydraulic conductivity of the soil near the surface in the grasslands areas of the Studibach ($5 \times 10^{-7}$ to $1 \times 10^{-5}$ m/s) but is not unrealistic for the forest sites ($>1 \times 10^{-4}$ m/s; van Meerveld et al. (2017)). For comparison, Anderson et al. (2009) determined subsurface velocities of $10^{-4}$ m/s (and up to $10^{-1}$ m/s) for preferential flow pathways in forest soils, whereas Uchida et al. (2001) mention velocities of $5 \times 10^{-3}$ m/s (and up to $2 \times 10^{-1}$ m/s) for pipeflow in forest soils. Most of the flow in the clay soils of the Studibach occurs through preferential flow pathways in the topsoil layers. However, as discussed on P6L1-12, the flowing stream network wouldn't be fully extended for many days in a row and the velocities will decrease as the catchment dries out. Therefore, in reality the travel times will be much longer than shown in Figure 4 (see Figure 6b).
>
> We do not know the travel time distribution for the catchment but the streamflow response in the catchment is very flashy. Previous studies have shown that the event water contributions to streamflow in the Alptal catchments can be very large (Fischer et al., 2017; von Freyberg et al., 2018b) and that the young water fraction can be very high (von Freyberg et al., 2018a).

7. It is also interesting that the median travel time for the topographic map survey is 4.5days and the subsurface travel time is 4.5 days, which are both longer than the "dry" conditions survey, and yet the fraction of the catchment with travel time less than 1 day is greater for the topographic map. Perhaps this is driven by the hydrologic connectivity of the river network in the topographic map survey. This is an interesting dynamic that could be expanded on in this paper and could be related to recent papers on the topic of discontinuous network extents, such as:

Godsey, S. E., & Kirchner, J. W. (2014). Dynamic, discontinuous stream networks:hydrologically driven variations in active drainage density, flowing channels and stream order. Hydrological Processes, 28(23), 5791-5803.
Whiting, J. A., & Godsey, S. E. (2016). Discontinuous headwater stream networks with stable flowheads, Salmon River basin, Idaho. Hydrological Processes, 30(13), 2305-2316.

> It is indeed interesting that the stream length and median travel time for the flowing stream network during dry conditions and the network from the topographic map are rather similar but the connected stream length (Table 1) and the area that likely contributes to the stream are very different (Figure 4). We already highlight this on P5L5-8. We also highlight the effect of the dry section in the flowing stream network on the travel time distribution and the area with travel times shorter than two days on P5L9-13.
>
> We already reference the mentioned publications but will include more information on them.

8. FIGURE 2: What is the role of disconnected stream channels in the model results? Do water parcels flow through these disconnected sections at the same rate as those coming from the terrestrial

landscape outside the channel extent? Do the authors think that subsurface flow map be faster within the subsurface channel network than in the hillslopes adjacent to the network?

> The disconnected section causes the second peak in the travel time distribution (see P5L9-13).

> We only used one subsurface flow velocity in our calculations. We agree that the flow may be faster through the channel bed than on the hillslopes but adding a different velocity for the area around the channel will make the results less clear. As mentioned throughout the text, the velocities were kept constant in order to avoid blurring the effect of the change in the stream network geometry on travel times by having different velocities. Of course in reality there will be a distribution of velocities, rather than one velocity for the entire catchment, and this velocity distribution will change as the catchment wets up or dries out. We describe this limitation and the effects that this has on the travel times on P5L24-31.

Technical corrections and editorial suggestions

P 5 L 11: missing "x" between "5" and "10".

> We will add a · between the 5 and the 10 (here and elsewhere in the text).

P 6 L 13: Delete "did" at end of sentence.

> We will remove the "did"

TABLE 2: Change "travel times smaller than one and two days" to "travel times shorter than one and two days"

> We will change the text of the caption accordingly.

**References:**

Anderson, A. E., Weiler, M., Alila, Y., and Hudson, R. O.: Subsurface flow velocities in a hillslope with lateral preferential flow, Water Resour. Res., 45, W11407, doi:11410.11029/12008WR007121, 2009.

Fischer, B. M. C., Stähli, M., and Seibert, J.: Pre-event water contributions to runoff events of different magnitude in pre-alpine headwaters, Hydrol. Res., 48, 28-47, 10.2166/nh.2016.176, 2017.

Uchida, T., Kosugi, K., and Mizuyama, T.: Effects of pipeflow on hydrological process and its relation to landslide: a review of pipeflow studies in forested headwater catchments, Hydrological Processes, 15, 2151-2174, 2001.

van Meerveld, H. J. I., Fischer, B. M. C., Rinderer, M., Stähli, M., and Seibert, J.: Runoff generation in a pre-alpine catchment: A discussion between a tracer and a shallow groundwater hydrologist, https://publicaciones.unirioja.es/ojs/index.php/cig/article/view/3349, 10.18172/cig.3349, 2017.

von Freyberg, J., Allen, S. T., Seeger, S., Weiler, M., and Kirchner, J. W.: Sensitivity of young water fractions to hydro-climatic forcing and landscape properties across 22 Swiss catchments, Hydrol. Earth Syst. Sci., 22, 3841-3861, 10.5194/hess-22-3841-2018, 2018a.

von Freyberg, J., Studer, B., Rinderer, M., and Kirchner, J. W.: Studying catchment storm response using event- and pre-event-water volumes as fractions of precipitation rather than discharge, Hydrol. Earth Syst. Sci., 22, 5847-5865, 10.5194/hess-22-5847-2018, 2018b.

---

## Author Comment (AC2) · 9 Aug 2019

Thank you very much for pointing us to this very interesting paper. We were not aware of it and agree that it is very relevant. We find it indeed very interesting that, although this is a very different system, the results are similar (and even the optimized velocities match the ones used in our study). Figure 7 in your manuscript clearly shows the importance of using a dynamic network for simulating the hydrograph. The difference in the optimized interfluve velocities for the conservative and non-conservative networks is exactly what we eluded to in the discussion of our manuscript (P6L25), where we

describe the importance of using dynamic stream networks for solute transport modeling because the use of a static network (as shown on maps) would lead to "slower modeled transport of pollutants, unless compensated otherwise (e.g. via velocities that are unrealistically high or large areas with surface runoff)".

Thank you for pointing us to this interesting paper. We will certainly reference it in our revised manuscript.

—————————————————————

---

## Referee Comment (RC2) · Anonymous Referee #2 · 16 Aug 2019

General Comments: This manuscript uses field-mapped stream extent and flow-routing from a digital elevation model to derive travel time distributions considering varying extents of the flowing stream network. The dynamic expansion and contraction of the stream network is not typically considered in this type of work. The manuscript makes a strong case for the acknowledgement of these processes in future travel time distribution work. I think the analysis is elegant and compelling and the manuscript is very well written. I have just a few questions and potential wording issues, which are noted below.

[Figure]

Specific Comments: Page 6, Line 13: "in our study did ." I can't quite figure out what this means, it may need to be reworded.

Figure 2: Definitely not critical, but it could offer helpful context to note the elevations of the lowest and highest contours in one of the maps.

Figure 4: I had a hard time interpreting the pie charts. From reading the caption, it seems like the blue in the pie chart represents the portion of the catchment sourcing water to the stream in 0-2 days (I think?). But then it doesn't seem like the pie charts match up with the corresponding maps. Are they somehow mismatched? If not, I'd suggest being more explicit what the pie charts represent. Another suggestion: I think they would be clearer just from a visualization perspective if instead of pies, they were rectangles... kind of like a progress bar on a computer. I think these would be easier to read and compare than the pie.

---

## Author Comment (AC3) · 2 Sep 2019

**Response to review comments from Anonymous Referee #2**

We appreciate the comments by the reviewer and the positive assessment of the manuscript. Below we respond (in blue text) to the individual comments (in black text).

General Comments: This manuscript uses field-mapped stream extent and flow-routing from a digital elevation model to derive travel time distributions considering varying extents of the flowing stream network. The dynamic expansion and contraction of the stream network is not typically considered in this type of work. The manuscript makes a strong case for the acknowledgement of these processes in future travel time distribution work. I think the analysis is elegant and compelling and the manuscript is very well written. I have just a few questions and potential wording issues, which are noted below.

Thank you for these positive comments.

Specific Comments:

Page 6, Line 13: "in our study did ." I can't quite figure out what this means, it may need to be reworded.

We agree that this sentence wasn't very clear. We meant to say that the travel times in the referenced studies were much longer than our calculated travel times (shown in Figures 4-6). We will rewrite this sentence.

Figure 2: Definitely not critical, but it could offer helpful context to note the elevations of the lowest and highest contours in one of the maps.

We will add the lowest and highest elevations to the map.

Figure 4: I had a hard time interpreting the pie charts. From reading the caption, it seems like the blue in the pie chart represents the portion of the catchment sourcing water to the stream in 0-2 days (I think?). But then it doesn't seem like the pie charts match up with the corresponding maps. Are they somehow mismatched? If not, I'd suggest being more explicit what the pie charts represent. Another suggestion: I think they would be clearer just from a visualization perspective if instead of pies, they were rectangles...kind of like a progress bar on a computer. I think these would be easier to read and compare than the pie.

Thank you for pointing us to this issue. Unfortunately, the pie charts changed when the document was converted to a pdf. The white part of the pie chart became blue, the darkest blue part of the pie chart disappeared, and the blue parts became white. This of course made it difficult to interpret the pie charts and caused the mismatch of the pie charts and the maps. We agree that a bar chart could also be nice but the space in the figure is limited and better suited to a pie chart.

We will export the figure differently and double check that pdf displays the figure correctly (see the figure below for the correct pie charts).

[Figure]

---

## Author Response (AR1)

We appreciate the comments by the reviewer and the generally positive assessment of the manuscript. Below we respond (in blue text) to the individual comments (in black text). The comments have helped us to clarify the manuscript.

**Response to reviewer 1 comments**

General comments
This article presents an interesting thought experiment about how riverine network length can influence the mean travel time distribution in catchments. The authors present a set of feasible river network extents across a range of wetness conditions, assume surface and subsurface flow velocities, and then estimate plausible distributions of travel times to the catchment outlet within these wetness scenarios. As this study is an initial exploration of how network extent can influence travel time distributions and modeling solute transport, I believe this study would be more powerful if the authors emphasized how future studies can build off of this initial exploration. For instance, emphasizing what the limitations of this study design are, and how others can use these concepts and apply real datasets and hydrologic measurements to confirm the results and interpretations of this study, would be greatly beneficial.

> We describe the limitations of the study (particularly the uniform and constant velocities and the 'steady state assumption' for each stream network) on P4L19-24 (last part of the methods) and P5L29-P6L5 (first part of the discussion). Our main goal was to show that the geometry of the flowing stream network affects the distribution of the hillslope travel distances to the flowing streams (and to a much smaller extent the travel distances in the stream) and thus the travel time distribution. We describe in the discussion what these results mean for interpreting travel time distributions obtained from tracer data and highlight that these results should be considered in solute transport models that - so far - tend to use a fixed (rather than dynamic) stream network.

Since this study estimates subsurface and surface velocities, it seems appropriate to provide results from a sensitivity analysis or provide ranges in the mean travel times. While the authors state they tested surface to subsurface velocity ratios (from 10 to 10000; P 4 L 14), they do not appear to present the results of that analysis. A powerful addition to this paper would be to show possible ranges in mean transit time distributions, given minimum and maximum velocities.

> We provide the results for different velocities in Figure 6 and describe them in the last paragraph of section 3. In short, the chosen velocities greatly affect the mean travel times and the range of travel times but have a minor effect on the shape of the travel time distribution or the differences in the travel time distributions for the different stream networks. Thus the main result of this study (namely, that the geometry of the flowing stream network affects the shape of the travel time distributions) does not depend on our chosen velocities. See also the response to specific comment 6 below.

Specific comments
1. P 3 L 26: How substantial of a rainfall event? Is the rainfall occurring in "wet" conditions? I suspect not as it occurs right before the "dry" conditions survey.

> There were 27 mm of precipitation on October 25$^{th}$ and another 31 mm fell on October 26th. There was no other rain after this event until November 2$^{nd}$ (total rainfall between October 15$^{th}$ and November 2$^{nd}$ was 83 mm). Streamflow in the catchment responds quickly to precipitation (within minutes to hours) and baseflow is generally reached within one to two days after an event. Thus, by November 2$^{nd}$, streamflow had returned to baseflow conditions, although the lowest flows during this fall period were increasing slightly (see Figure R1 below).

[Figure]

Figure R1. Daily precipitation and streamflow at the Erlenbach gauging station during spring to-fall 2016. The red lines indicate the times of the two stream surveys in fall 2016. Note the log-scale for streamflow. The flow during the extremely dry conditions in summer 2018 was 0.18 mm/d. The data were obtained from Stähli (2018), Long-term hydrological observatory Alptal (central Switzerland); https://www.envidat.ch/dataset/longterm-hydrological-observatory-alptal-central-switzerland.

2. P 3 L 20: The authors say that the field mapping is too slow during rainfall events to capture the entire extent of the stream work during rainfall events due to how dynamic it is. However, it appears the authors use a survey taken during a rainfall event in this analysis. Thus, it would be helpful to the reader if more information was provided on these surveys, e.g. how long did the surveys take, did the researchers start at the channel heads and walk down (to ensure they capture the most dynamic extents), was the network actively expanding during the survey, etc?

We didn't survey the stream starting at the channel heads but rather walked along the contours and then up the different streams (thus the surveys were done in more of a zigzag pattern across the catchment). Each survey took at least half a day to complete. Since the peak of the event is very short, we cannot survey the entire stream network at the peak of the event. We made it clearer in the text of the manuscript that the mapping is too slow to capture the peak flow conditions during an event (P3L20-21).

The October 2016 stream network was mapped in the afternoon of October 25[th] during an event with low intensity rainfall (see Figure R2 below). Total rainfall was 10 mm by noon (when the mapping started), 16 mm by 3 pm, and 20 mm at 5 pm when the mapping was completed.

[Figure]

Figure R2. Hourly and cumulative precipitation recorded at the lower and upper rain gauge in the Studibach before and during the stream survey on October 25, 2016.

3. P 3 L 27: It may be more clear to the reader how survey #4 was accomplished (every other survey is described in parentheses, but this one).

> The stream network was surveyed on multiple occasions to ensure that we had mapped all streams. The complete network is assumed to represent the fully extended network during extremely wet or peak flow conditions. We did not observe flow in all streams at the same time; instead, #4 represents a hypothetical scenario in which all channels are flowing during extremely wet conditions. We explain this in the text on P3 and added a clarifying sentence to the caption of Table 1 that the complete network is assumed to represent the flowing stream network during extremely wet conditions.

4. There have been several recent studies that sought to predict river network extent, which can be used to model transit time distributions as suggested by the authors on
P33. Some suggestions below for two recent studies that can be used:
P 7 L 1: Add another example of predictive modelling: Ward, A. S., Schmadel, N. M., & Wondzell, S. M. (2018). Simulation of dynamic expansion, contraction, and connectivity in a mountain stream network. Advances in Water Resources, 114, 64-82.
P 6 L 33- P 7 L 1: Add example of empirical generalization from field studies, such as: Zimmer, M. A., & McGlynn, B. L. (2018). Lateral, vertical, and longitudinal source area connectivity drive runoff and carbon export across watershed scales. Water Resources Research, 54(3), 1576-1598.
This study also relates network expansion and retraction to solute transport dynamics as well, which is suggested in this study, but few if any citations are provided.

> Thank you for these suggestions. We have added references to these papers and other stream network model studies to the manuscript (P6L29 and P7L8).

5. TABLE 1: While it is clear why the topographic map does not have an associated streamflow, please add brief explanation in caption as to why streamflow magnitude is not provided for complete network.

> We added the clarification to the table caption and now clearly state that we never observed flow in the entire stream network but that we assume that this is the case for peak flow conditions during very large events.

6. TABLE 2: This is an incredibly interesting results table and definitely made me think about possible travel times in other catchments and across wetness conditions. While I think the authors main

points from this paper were to show that travel times decrease substantially as the system wets up, the absolute values for the reported median travel times are very small. The median surface travel times are on the order of minutes – how did the authors determine this? Based on the catchment and previous field observations, does it seem reasonable that 71% of the water travel time are less than 2 days?

> The main point of the study was indeed to show that the changes in the flowing stream network geometry can significantly affect the travel time distribution. We did not intend to derive actual travel time distributions for this catchment but rather focus on how the distributions differ between the different stream networks.
>
> We don't know the average surface and subsurface velocities in the catchment and therefore state clearly that these are assumed values. We test the effect of using different velocities and show that the effect of the chosen velocities on the shape of the travel time distribution is minimal (see Figure 6 in the manuscript). Furthermore, we discuss on P5L29-P6L5 the implications of using uniform and constant velocities.
>
> The average surface velocity of 0.5 m/s is typical for mountain streams. The average subsurface velocity of $5 \ 10^{-4}$ m/s is high compared to the hydraulic conductivity of the soil near the surface in the grassland areas of the Studibach ($5 \ 10^{-7}$ to $1 \ 10^{-5}$ m/s) but is not unrealistic for the forest sites ($>1 \ 10^{-4}$ m/s; van Meerveld et al. (2017)). For comparison, Anderson et al. (2009) determined subsurface velocities of $10^{-4}$ m/s (and up to $10^{-1}$ m/s) for preferential flow pathways in forest soils, whereas Uchida et al. (2001) mention velocities of $5 \ 10^{-3}$ m/s (and up to $2 \ 10^{-1}$ m/s) for pipeflow in forest soils. Most of the flow in the clay soils of the Studibach occurs through preferential flow pathways in the topsoil layers. However, as discussed on P4L19-27 and P6L1-5, the flowing stream network wouldn't be fully extended for many consecutive days and the velocities will decrease as the catchment dries out. Therefore, in reality the travel times will be much longer than shown in Figure 4 (see Figure 6b).
>
> We do not know the travel time distribution for the catchment but the streamflow response in the catchment is very flashy. Previous studies have shown that the event water contributions to streamflow in the Alptal catchments can be very large (Fischer et al., 2017; von Freyberg et al., 2018b) and that the young water fraction can be very high (von Freyberg et al., 2018a).

7. It is also interesting that the median travel time for the topographic map survey is 4.5days and the subsurface travel time is 4.5 days, which are both longer than the "dry" conditions survey, and yet the fraction of the catchment with travel time less than 1 day is greater for the topographic map. Perhaps this is driven by the hydrologic connectivity of the river network in the topographic map survey. This is an interesting dynamic that could be expanded on in this paper and could be related to recent papers on the topic of discontinuous network extents, such as:

Godsey, S. E., & Kirchner, J. W. (2014). Dynamic, discontinuous stream networks:hydrologically driven variations in active drainage density, flowing channels and stream order. Hydrological Processes, 28(23), 5791-5803.

Whiting, J. A., & Godsey, S. E. (2016). Discontinuous headwater stream networks with stable flowheads, Salmon River basin, Idaho. Hydrological Processes, 30(13), 2305-2316.

> It is indeed fascinating that the stream length and median travel time for the flowing stream network during dry conditions and the network from the topographic map are somewhat similar but the connected stream length (Table 1) and the area that likely contributes to the stream are very different (Figure 4). We highlight this on P5L10-13. We also highlight the effect of the dry section in the flowing stream network on the travel time distribution and the area with travel times shorter than two days on P5L14-18.
>
> We referenced the mentioned publications but now added references where we describe the discontinuity in the flowing streamwork as well (P4L1-2).

8. FIGURE 2: What is the role of disconnected stream channels in the model results? Do water parcels flow through these disconnected sections at the same rate as those coming from the terrestrial landscape outside the channel extent? Do the authors think that subsurface flow map be faster within the subsurface channel network than in the hillslopes adjacent to the network?

> The disconnected section causes the second peak in the travel time distribution (see P5L14-18).
>
> We only used one subsurface flow velocity in our calculations. We agree that the flow may be faster through the channel bed than on the hillslopes but adding a different velocity for the area around the channel will make the results less clear. As mentioned throughout the text, the velocities were kept constant in order to avoid blurring the effect of the change in the stream network geometry on travel times by having different velocities. Of course in reality there will be a distribution of velocities, rather than one velocity for the entire catchment, and this velocity distribution will change as the catchment wets up or dries out. We describe this limitation and the effects that this has on the travel times on P5L31-P6L2.

Technical corrections and editorial suggestions

P 5 L 11: missing "x" between "5" and "10".

> We added a · between the 5 and the 10 (here and elsewhere in the text).

P 6 L 13: Delete "did" at end of sentence.

> We removed the "did" and rewrote the sentence based on the comments from reviewer 2.

TABLE 2: Change "travel times smaller than one and two days" to "travel times shorter than one and two days"

> We changed the text of the caption accordingly.

**Response to review comments from Anonymous Referee #2**

General Comments: This manuscript uses field-mapped stream extent and flow-routing from a digital elevation model to derive travel time distributions considering varying extents of the flowing stream network. The dynamic expansion and contraction of the stream network is not typically considered in this type of work. The manuscript makes a strong case for the acknowledgement of these processes in future travel time distribution work. I think the analysis is elegant and compelling and the manuscript is very well written. I have just a few questions and potential wording issues, which are noted below.

      Thank you for these positive comments.

Specific Comments:

Page 6, Line 13: "in our study did ." I can't quite figure out what this means, it may need to be reworded.

      We agree that this sentence was awkward. We meant to say that the travel times in the referenced studies were much longer than our calculated travel times (shown in Figures 4-6). We have rewritten this sentence.

Figure 2: Definitely not critical, but it could offer helpful context to note the elevations of the lowest and highest contours in one of the maps.

      We highlighted the 50 m contours as well and added the elevations to these contours in Fig 2e. Note that we give the elevation range on P3L6.

Figure 4: I had a hard time interpreting the pie charts. From reading the caption, it seems like the blue in the pie chart represents the portion of the catchment sourcing water to the stream in 0-2 days (I think?). But then it doesn't seem like the pie charts match up with the corresponding maps. Are they somehow mismatched? If not, I'd suggest being more explicit what the pie charts represent. Another suggestion: I think they would be clearer just from a visualization perspective if instead of pies, they were rectangles...kind of like a progress bar on a computer. I think these would be easier to read and compare than the pie.

      Thank you for pointing us to this issue. Unfortunately, the colors in the pie charts changed when the document was converted to a pdf(the white part of the pie chart became blue, the darkest blue part of the pie chart disappeared, and the blue parts became white). This of course made it difficult to interpret the pie charts and caused the mismatch of the pie charts and the maps. We agree that a bar chart could also be nice but the space in the figure is limited and better suited to a pie chart.

      We exported the figure differently and double checked that the pdf displays the figure correctly (see the figure below for the correct pie charts).

[Figure]

**Response to comment by Kang Yang**

A very interesting study! It has been long known that hillslope and open-channel partition impacts catchment hydrographs but few studies have estimate this point using high-quality dataset. The main contribution of this study is to map multi-temporal, accurate flowing stream networks and investigate their impacts on catchment hydrographs. Some important implications have been reported as well. This reminds me of the work I've done for routing surface meltwater on the Greenland ice sheet. I've found hillslope and open-channel partition impacts surface meltwater discharge at the catchment outlet. However, I used a series of cumulative area thresholds to create dynamic supraglacial stream networks from DEMs (see Figure 7). This study has done a better work: instead of using DEM simulations, real field-measured stream networks are used. If the authors are interested, see my paper published in The Cryosphere:

Yang, K., Smith, L.C., Karlstrom, L., Cooper, M.G., Tedesco, M., As, D.v., Cheng, X., Chen, Z., Li, M., 2018. A new surface meltwater routing model for use on the Greenland Ice Sheet surface. Cryosph. 12, 3791-3811. https://www.the-cryosphere.net/12/3791/2018/

> Thank you very much for pointing us to this interesting paper. We were not aware of it and agree that it is relevant. We find it indeed very interesting that, although this is a very different system, the results are similar (and even the optimized velocities match the ones used in our study). Figure 7 in your manuscript clearly shows the importance of using a dynamic network for simulating the hydrograph. The difference in the optimized interfluve velocities for the conservative and non-conservative networks is precisely what we alluded to in the discussion of our manuscript (P6L30), where we describe the importance of using dynamic stream networks for solute transport modeling because the use of a static network (as shown on maps) would lead to "slower modeled transport of pollutants, unless compensated otherwise (e.g. via velocities that are unrealistically high or large areas with surface runoff)".
> Thank you for pointing us to this interesting paper. We now reference it in our revised manuscript (P6L31).

[revised manuscript text omitted]